# The Perception of the Impact of Land-Use on Small and Large Cities by Tourists Using p2p Platforms

Sara Calvo [1,*], Andres Morales [1], Miguel Angel Del Arco [1] and Gualter Manuel Medeiros do Couto [2]

1    Social Sciencies and Business School, Universidad Internacional de La Rioja, Avenida de la Paz, 137,
     26004 Logroño, Spain
2    School of Business and Economics and CEEAplA, University of Azores, 9500-321 Ponta Delgada, Portugal
*    Correspondence: sara.calvo@unir.net

**Abstract:** In the literature, numerous impacts on local communities associated with the activity of online platforms for the intermediation of tourist accommodation have been described. Previous studies analyzed the perception associated with some of these impacts; however, the importance attributed by tourists using p2p platforms to the impact of the use of land has not been analyzed. Moreover, few studies have explored the different perceptions of these impacts on small and large cities. Therefore, this work, based on the information provided by 294 surveys associated with an online MOOC, analyzes the perception of users (tourists) of these p2p platforms on the impact of the use of land in small and large cities. The results have demonstrated that there is a great difference between the perception of respondents on this impact, with a worse perception of the use of p2p platforms by tourists who live in cities with a greater number of inhabitants.

**Keywords:** tourism; land-use impact; p2p; small and large cities





## 1. Introduction

There are several phenomena that have significantly affected the social sustainability of communities receiving tourism in the past decade. Among them, the massive arrival of tourists stands out as a consequence of decades of pro-growth policies and the expansion of low-cost tourism [1,2]. To this is added the low seasonality described for urban destinations, which implies that there are practically no rest periods for residents [3]. An additional factor is the exponential growth in the supply of tourist accommodation located in residential buildings [4,5]. This offer is mainly linked to homes brokered on online platforms. The sum of these factors has led to a growing feeling of tourism-phobia, even in environments where economic activity depends to a large extent on tourism [2]. Thus, the concepts of the capacity of destinations and over-tourism have taken on a new relevance, since a social problem has been generated with a growing impact [6]. Moreover, in recent years, there has been an increment in tourism in rural areas and small cities, particularly since COVID-19 [7,8]. Focusing attention on the object of study of this work, the expansion of online platforms for the intermediation of tourist accommodation is clearly verified. The activity of p2p platforms for the intermediation of tourist accommodation began in 2008 and since then its growth has been intense [9]. The leading company in the p2p sector is Airbnb, although many others operate such as Wimdu, 9flats, Roomorama, Onefinestay, HouseTrip, FlipKey, CouchSurfing, and MisterAirbnb. These companies can be defined as "internet-based companies that allow ordinary people to offer tourist accommodation" [10,11]. Only taking data from Airbnb, it is verified that it offers more than 5.6 million accommodations in more than 200 countries and 100,000 cities [12]. This has undoubtedly generated interference with local communities [3], apart from debates about its legitimacy [6,7] mand the direction that regulation should take [5,9].

Tourism activity generates effects, both positive and negative, on the land and communities in which it takes place [5,13]. These effects are known as land use impacts, and

can be classified as economic, social, labor and environmental [14]. Since its appearance in 2008, the impact of land use associated with online tourist accommodation brokerage platforms has been widely studied [14,15]. Within this subject there is a more specific one, linked to the perception of this type of impact by the affected groups [16]. Studies have been published focusing on the perception linked to specific impacts by the main groups affected. Although up to now no international studies have been carried out to analyze the perception of the tourists themselves, or consumers of this type of accommodation service, and of the set of impacts associated with this type of accommodation in small and large cities, which is the objective of this work. This type of analysis is highly relevant. In the first place, several authors point out that the academic literature on this subject is scarce and that complementary analyzes are required [17,18]. In addition, perception analyzes are necessary in order to regulate an activity that is poorly legislated, which in itself is perceived as one of the triggers for social rejection [3].

The proposed research is based on what has been previously described and focuses its contribution on two research questions. RQ1: Are there differences between the way tourists perceive the impact of land use linked to these accommodations in small and large cities? RQ2: Which effects are perceived to a greater extent? The study is based on the responses issued in the framework of a survey answered by students of a MOOC presented internationally. In this paper, we classify small cities as those that have 250,000 or less inhabitants. This MOOC is titled Social Innovation: Global Solutions for a Sustainable Future and is available on the FutureLearn platform. A total of 294 individuals responded to the survey between the months of December 2021 and November 2022. Citizens from 48 countries were included in the sample, which provided a valuable international perspective. This undoubtedly provides added value to this research, as it allows us to determine the global perception of this type of activity.

## 2. Land Use Impacts Linked to the Consumption of Accommodation Services Located in Residential Environments

Tourism activity has the potential to contribute to socio-economic development in a sustainable way, but it can also generate negative impacts on destinations [19,20]. The final balance depends on numerous factors, such as the design of the destination itself, the volume of tourists arriving, the type of activities carried out, the environmental and social fragility of the surroundings, the degree of seasonal concentration of flows and the regulation that orders tourist activity [21]. The opinion of the stakeholders affected by tourist activity must be taken into account in regulation and planning processes [5,22,23], while the viability of a destination will be reduced without the support of the locals [3].

The land use impacts described below focus on the activity of tourist accommodation brokerage platforms. New types of impacts associated with this accommodation option have been generated or, more precisely, with the increase in the density of tourist accommodation in residential settings fostered by p2p platforms [24,25]. The p2p platforms have intensified an activity that is not new, since tourist homes have operated in the sector on a regular basis [26]. Some of these impacts are variations of others previously described. The rapid expansion of the offer of this type of accommodation has caused impacts in the traditional tourist accommodation sector, but also in residential neighborhoods. It is possible to classify the land use impacts generated in the following categories: sociocultural, economic, labor and environmental impacts [3]. The increase in the price of housing, which is an economic impact, generates a loss of population and cohesion in neighborhoods [9,11,27].

The economic impacts associated with these platforms have aroused much controversy, partly as a consequence of the fact that the final balance is difficult to determine [28]. The controversy is also associated with a debate on the legitimacy of this accommodation offer [6,7]. Various positive impacts of an economic nature have been described. This modality of accommodation makes it possible to offer places at a lower price [14,27,29], which permits tourism for a segment of the population that could not have enjoyed it otherwise [30]. The offer of accommodation places offered by these houses complements

the traditional form, which increases the capacity of the destinations and makes it possible to take advantage of peak times [27,29]. Even though these homes are usually highly concentrated in tourist areas, they are also located outside them, which allows visitor spending to occur in different areas to the central ones [31]. In fact, in some cases the effect of these houses on the revitalization of neighborhoods has been described [19]. The alteration of consumption patterns has benefited restaurants and shops, since visitors spend more in them to the detriment of hotels [32]. For many entrepreneurs, it is easier to start their projects within the framework of the collaborative economy [33–35]. The economic benefits for homeowners are clear, as the additional income supplements the main one and helps pay mortgages [23,30]. To these positive impacts are added others of a less friendly nature. A large part of tourist housing is in the hands of investment groups, which limits the ability of this activity to benefit a large number of citizens [26]. One of the most widespread impacts derives from the effect on the housing market, which has generated a shortage in the supply of long-term rental housing and has generated evictions, while the activity of these platforms addresses access to housing for residential use, [17,18]. This supply of accommodation, sometimes as a consequence of weak regulation, is associated with the informal economy, unfair competition and tax evasion [25,36,37], although it has also been pointed out that p2p platforms have brought to light undeclared offers [38]. Of course, studies have been described, although not conclusive, on the effect generated in the traditional accommodation offer [10].

Once the main economic impacts have been described, which partly condition the socio-cultural impacts, progress can be made in presenting the latter. This assumes the potential effect that the expansion of this type of accommodation can generate on the lives of residents in tourist areas [9] and on the loss of quality of life [24]. The excessive focus on tourism activities that is taking place in many traditionally residential areas can alter the cohesion of communities and damage local culture [11,39]. The alteration in the lives of residents can derive from the effects associated with increased traffic, increased noise in residential buildings and congestion in public spaces [9,23,26,35]. A substitution of traditional commercial activities with others focused on tourists has been observed [25,36]. The type of tourism that these tourist homes can attract has been related to an increase in the consumption of alcohol and drugs in these residential settings, as well as problems of insecurity for residents [9,26]. Security problems also refer to tourists, as they might stay in poorly regulated and professionalized environments [40], in which security in the transaction is not guaranteed, nor are hygiene standards. Returning to the problem of the loss of cohesion in neighborhoods, this problem is partly derived from the loss of residents, who are replaced by passing visitors [9,26]. Not all the impacts described within this dimension are negative; others have also been found that generate positive effects and lead to new opportunities for the local community. Such would be the case for the effect of the expansion of values associated with the collaborative economy, such as mutuality, equality, openness, honesty, an ethic of care and empathy [33] and collaboration between equals [4].

The effect described on the loss of cohesion can evolve towards an increase in neighborhood identity and promote actions that reinforce the cohesion that is in danger [34]. In Stephany's initial vision, the use of empty or underused rooms or homes implied a more efficient use of resources [41]. For tourists, these houses make it possible to enjoy a more authentic experience, since contact with local communities increases [1,20,40]. As part of the social impacts are focused on the labor market, there are many studies that warn of job insecurity associated with the collaborative or p2p economy [36,40], even when it implies a potential to complement income from main activities. In addition, the lack of regulations that are applied to these homes reduces the production costs associated with this accommodation offer, which ends up putting pressure on the reduction of labor costs associated with the traditional accommodation offer [42]. The lower occupancy rate in hotels has led to staff layoffs, which have not been compensated for by hiring a greater number of workers in other types of accommodation [19].

In the following section, we will explain the methodology used for this study. Few studies have explored the perception of land use impact in small and large cities [8,16,43]. For example, an interesting study conducted in Girona, a small city in Spain, highlighted that smaller tourism destinations might mirror themselves on those close to large tourism destinations, and thus residents' opinions and attitudes seem to be influenced by the situations experienced in these larger tourism destinations [43]. Another study conducted in Korea about trends in social networking services (SNS), which investigated the location selection of Food and Beverage (F&B) customers, showed that the place selection pattern via SNS was different spatially and quantitatively from that via the established retail location principle [35]. As such, the study revealed a novel behavior pattern regarding place selection based on SNS. Moreover, this fact indicates that the consideration of gradual changes in the selection of urban project sites, development of land use plans, and calculation of rent for retail is needed in light of advancements in SNS.

## 3. Methodology

In 2020, Minca Ventures Ltd.(London, United Kingdom) developed a massive social enterprise MOOC program titled *Social innovation: Global Solutions for a Sustainable World* on the FutureLearn platform, with the purpose of inspiring people to learn about social innovation and sustainability. The MOOC program consisted of three learning weeks (4 h per week with a total of 12 h). Each week contained a set of activities, which in turn contained a set of steps along with a set of learning activities. The steps consisted of discussions, social-media-related activities, video case studies, articles, quizzes, practical exercises, peer review and problem-based activities. In one of the steps in the last week of learning, a survey was included to investigate the perceptions of the learners as tourists on the land use impact associated with consumption via p2p platforms.

*Data Collection and Analysis: Cluster Analysis*

A survey of the participants was opened and available in the final steps of the three-week course. Out of 1364 active learners on the course, 294 people participated in the survey. The survey was implemented in English via Google Form between December 2021 and November 2022. The survey consisted of 37 questions, some of them descriptive (6 in total), 7 categorical (yes/no) and 24 used a Likert scale (1 = perception of low impact and 10 = perception of high impact) (see the survey questions in Table 1). Quantitative analysis involved descriptive statistics of the online survey results.

It is very important to highlight the difficulty of analyzing the land use impacts that have been described, separating the effect derived from over-tourism from the effects of the platforms themselves [14]. In this case, respondents were asked to focus on the effects associated with the activity of these platforms and their perception of these. It is observed that consumption patterns are increasingly influenced by the perceived environmental impact [15,44]. This work focused solely on revealing whether consumers from small and large cities perceive the land use impacts associated with their consumption differently. The role of this perception of their own values in the consumption decision was not analyzed. This is outside the focus of this research, since it would in itself be the continuation of the current research and should take up academically accepted frames of reference [45]. Classification algorithms run cluster analysis on a large data set to filter out data that belongs to obvious groups. In this work, in addition to descriptive results, some data were obtained through the performance of a cluster analysis. Cluster analysis is often used as a preprocessing step for various machine learning algorithms. It was deemed opportune to carry out this analysis in order to evaluate possible aspects that differentiate population groups by the size of the city [46].

**Table 1.** Key questions formulated in the survey.

| | | |
|---|---|---|
| 1. p2p platforms used by tourists | 2. When selecting accommodation, you give value to the economic impact on the community | 3. When selecting accommodation, you give value to the environmental impact generated |
| 4. When selecting an accommodation, you give value to the social impact generated | 5. Negative impact on social sustainability | 6. You believe tourism increases wealth |
| 7. You believe tourism increases tax collection | 8. You believe tourism brings greater opportunities for local entrepreneurs. | 9. You believe tourism brings an increase in employment rate |
| 10. You believe tourism brings an increase in the leisure offer | 11. You believe tourism brings an improvement in the state of conservation of buildings | 12. You believe tourism brings an increase in the value of business |
| 13. You believe tourism brings greater cultural interaction | 14. You believe tourism prevents the deterioration of urban and historical areas | 15. You believe tourism improves the image of the city |
| 16. You believe tourism increases the price of housing and business establishments | 17. You believe tourism increases prices | 18. You believe tourism causes a loss of local population |
| 19. Loss of traditional business establishments | 20. Transmission of diseases carried by tourists | 21. Increased insecurity; alcohol, drugs, etc. |
| 22. Negative effects on traditional accommodations: hotels, etc | 23. Increased congestion: public spaces, traffic, etc. | 24. Disruption of tranquility |
| 25. Loss of social cohesion | 26. Importance of tourist activity in your city | 27. In the building in which you reside, there is a tourist apartment |
| 28. Are you a tourist apartment owner? | 29. Do you work or own a business at street level in a tourist area? | 30. Age |
| 31. Level of studies completed | 32. Size of the city in which you reside | 33. Describe your city/town and country. |
| 34. Opinion on the effect that tourist housing rental platforms have on your environment | 35. Does your income depend in any way on the tourism sector? | 36 & 37. Aware through the media of possible positive and negative on tourism in the local community |

Source: compiled by the authors.

However, the choice of how to measure the distance between two elements is something that is not agreed upon in the literature and there are many subjective alternatives. Distance is a numerical measure of how far two individuals are apart; in other words, it measures the proximity or similarity between individuals. The most common way to measure distance is Euclidean distance, although there are other alternatives, such as the Manhattan distance used for particular types of problems. However, our data contains mixed data types (numerical and categorical) where these distances are not applicable, therefore traditional clustering algorithms such as K-means or hierarchical clustering are not valid. Therefore, for our study, we use the Gower distance, which is a measure of distance that can be calculated for two individuals whose attributes are mixed. The Gower distance is computed as the average of the dissimilarities between individuals. Each Gower distance lies between (0, 1).

$$d(i,j) = \frac{1}{p} \sum_{i=1}^{p} d_{if}^{(f)}$$

Source : compiled by the authors (1)

The partial dissimilarity $d(i,j)$ $(f)$ depends on the type of variable that we are measuring. In the case of numerical variables, partial dissimilarity is the ratio between the absolute differences between observations and the maximum observed range of all individuals. In the case of categorical variables, the partial dissimilarity is 1 if the observations are different and 0 if not. The selected clustering algorithm should fit well with the Gower distance. To do this, we selected the k-medoids algorithm. The k-medoid algorithm, Partitioning

Around Medoids (PAM), is a classic partitioning method similar to the well-known k-means method but, instead of iterating over the centroids, it iterates over the medoids, that is, it tries to find the most representative object for each cluster [46]. The algorithm clusters the objects in a total of k clusters, where k must be given a priori. The selection of the optimal number of clusters (k) must be made considering statistical information obtained in the data, although if there is any reasoned justification a priori, the number of clusters may vary for different reasons. To select the optimal number of clusters into which to divide our data, we use the silhouette width. The silhouette width is one of the most commonly used options for measuring the similarity between each point in a cluster and compares this similarity with the closest point of the neighboring cluster. This metric lies between (−1, 1) where higher values mean greater similarities. Figure 1 shows the result of the measurement for values of k between 2 and 10, where it can be observed that segmenting the students into 2 groups maximizes the similarity within the clusters and the dissimilarity between clusters. We have divided the sample into 2 groups following the results found in the silhouette analysis; however, using k = 2 produces useful results for grouping respondents. We use this approach to segment the respondents, leaving question P32 out of the analysis, which is that which defines the size of the city in which the respondent lives. In the following section we present the findings of the survey.

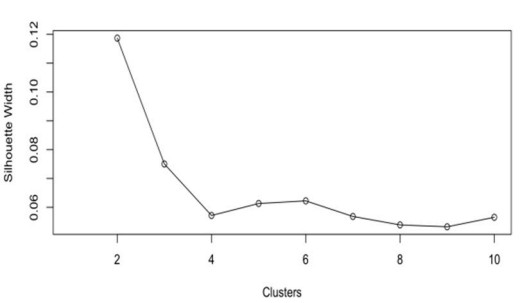
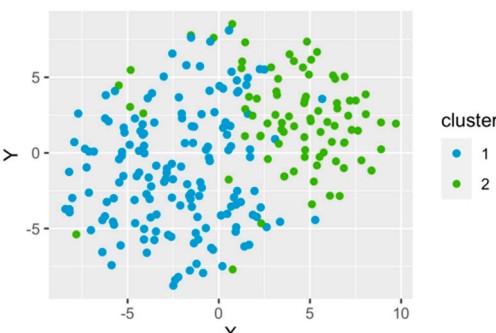

**Figure 1.** Optimal number of clusters with Silhouette Width and Cluster plot (k = 2). Source: compiled by the authors.

## 4. Results

In terms of the size of the city, there were 13.98% with fewer than 20,000 inhabitants, 7.34% with between 20,000 and 50,000 inhabitants, 20.62% with between 50,000 and 250,000 inhabitants, 23.07% with between 250,000 inhabitants and 1 million and 34.97% with more than 1 million inhabitants. As indicated in Figure 2, the majority of participants were between 35 and 55 years old (37.5%), followed by participants between 18 and 35 years old (34%). Most respondents had a university educational level (70%), with some of with secondary studies (24%) and a few with primary studies (6%). When asking respondents about their opinion on the effect that tourist housing rental platforms have on their environment, the majority of respondents suggested that the effect is neutral (43%), followed by a positive effect (24%). Only 8% of respondents claimed that tourism affected their environment very positively. A few respondents suggested that tourism affected their environment negatively (20%) and very negatively (5%) (see Figure 2 for details).

When asking respondents whether they had ever used the services of tourist accommodation platforms such as Airbnb, Wimdu, 9flats, Roomorama, Onefinestay, HouseTrip, FlipKey, CouchSurfing, and MisterAirbnb, 72% of respondents stated they had used them.

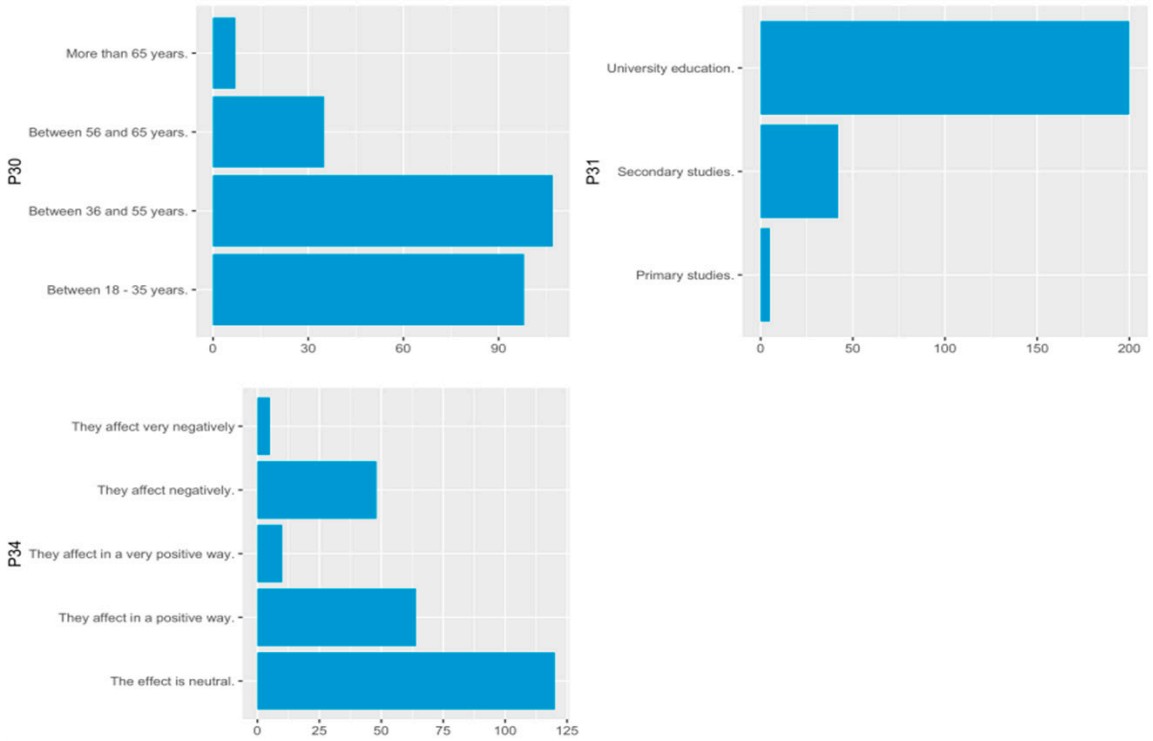

**Figure 2.** Summary of responses for age, education and opinion regarding the impact of tourism. Source: compiled by the authors.

Figure 3 shows that the majority of respondents did not participate in anything related to touristic activities. Another interesting finding emerged, as most respondents pointed out that they were aware through the media of possible negative effects on the receiving communities associated with the expansion of tourist housing, and only few on the possible positive effects. Figure 4 represents all the answers to the questions that are allocated within the Likert Scale (0–10). What can also be seen from the results is that the majority of respondents saw tourism as a great opportunity for local entrepreneurs (P8). There were some factors highlighted as impacting the local community negatively, in particular the increase in the price of housing and business establishments (rental and sale), the increase in prices in shops, bars, etc., and high levels of congestion in public spaces with more traffic. This is in line with previous studies that suggest negative impacts on the land use of tourism in the community [5,13–15].

*Cluster Differences*

This section presents the results of the k-medoids clustering technique using the Gower distance. As defined in the methodology section, the k-medoids machine-learning algorithm for clustering data defines the most representative object in each cluster, which, in our study, can be defined as the student profile in each cluster. Table 2 shows the characteristic that defines the profile of each cluster using k-medoids (K = 2). Furthermore, along with these characteristics, we present the conversion rate. What we found from the analysis is that there was a difference between clusters regarding the size of the city. As can be seen in cluster 1, the people who live in large cities, between 250,000 inhabitants and 1 million (70.18%) and more than a million (70.45%), were within Cluster 1.

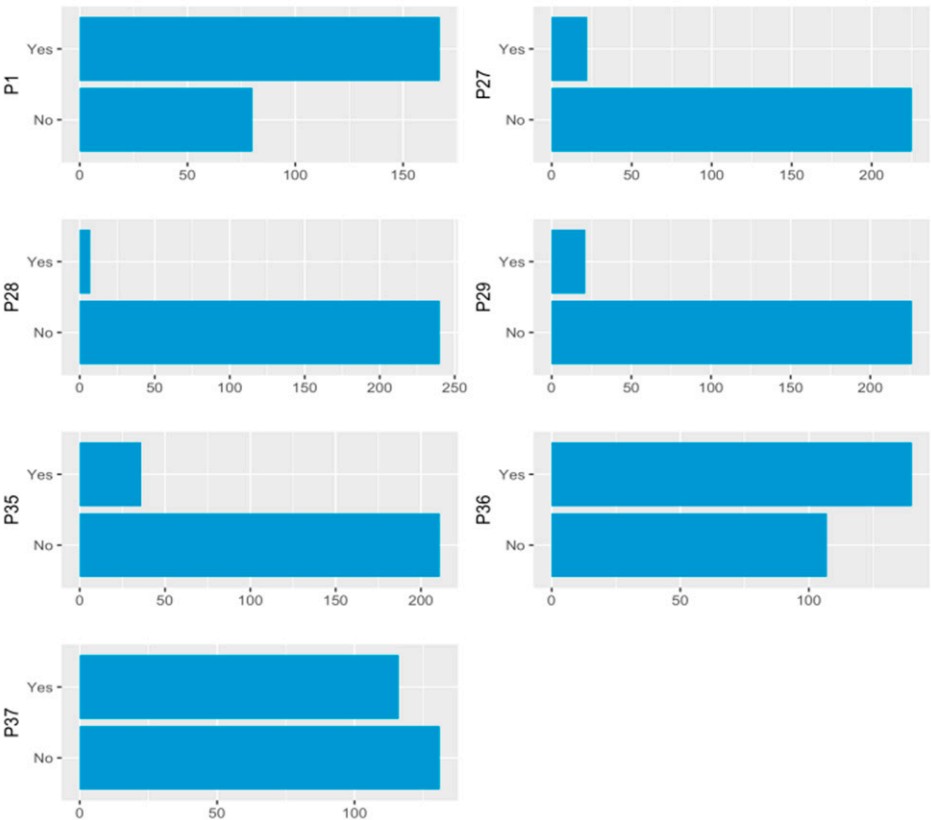

**Figure 3.** Summary of responses to questions with categorical responses from two categories. Source: compiled by the authors.

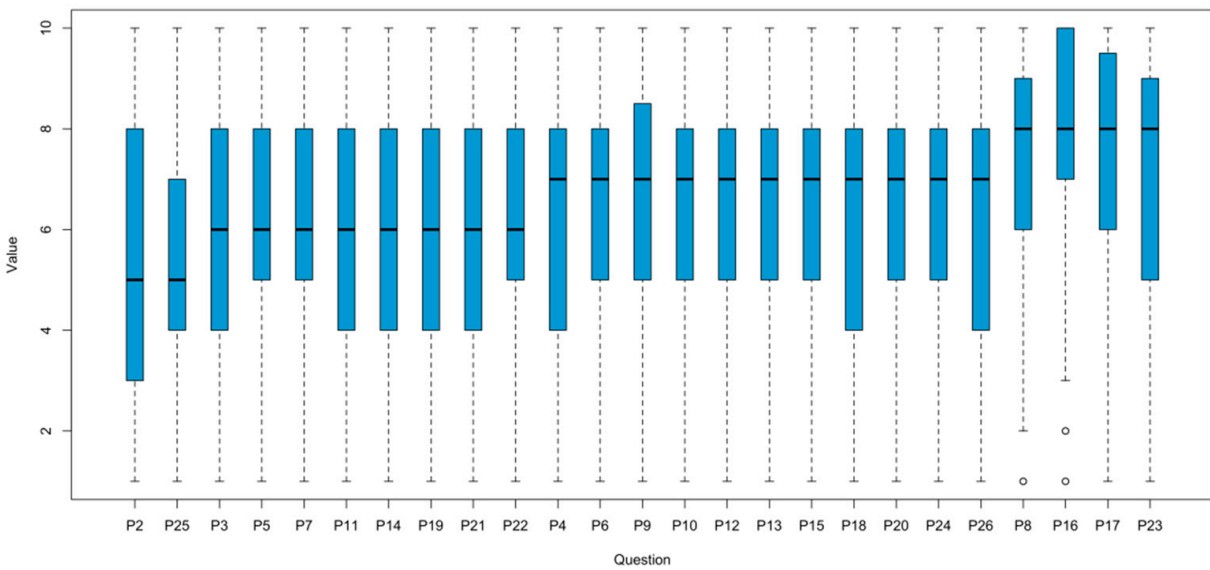

**Figure 4.** Summary of answers to the questions with quantitative answers (0–10). Source: compiled by the authors.

**Table 2.** Proportion of respondents from each cluster based on the size of the city in which they live.

| Size of the City | Cluster 1 | Cluster 2 |
|---|---|---|
| Less than 20,000 inhabitants | 56.25% | 43.75% |
| Between 20,000 and 50,000 inhabitants | 62.50% | 37.50% |
| Between 50,000 and 250,000 inhabitants | 53.70% | 46.30% |
| Between 250,000 inhabitants and 1 million | 70.18% | 29.82% |
| More than 1 million inhabitants | 70.45% | 29.55% |

Source: compiled by the authors.

For a better visualization of our results, we represent the variables in which we find differences between clusters, that can be considered as the main discriminatory variables between the two clusters in Figures 5–7. As seen in Figure 5, the vast majority of respondents have used the services of tourist accommodation platforms such as Airbnb, Wimdu, 9flats, Roomorama, Onefinestay, HouseTrip, FlipKey, CouchSurfing, and MisterAirbnb. Moreover, the vast majority of respondents did not have a touristic apartment or own a company that benefits from touristic activities (see Figure 5).

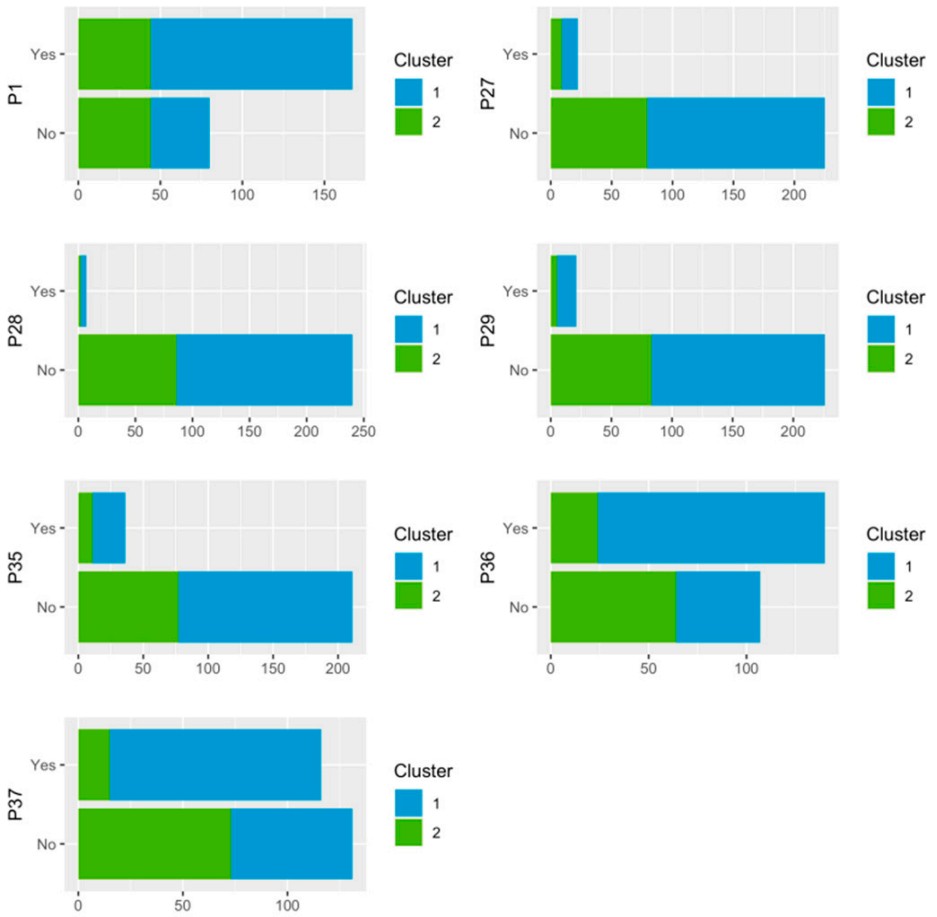

**Figure 5.** Summary of responses by cluster to questions with categorical responses from two categories. Source: compiled by the authors.

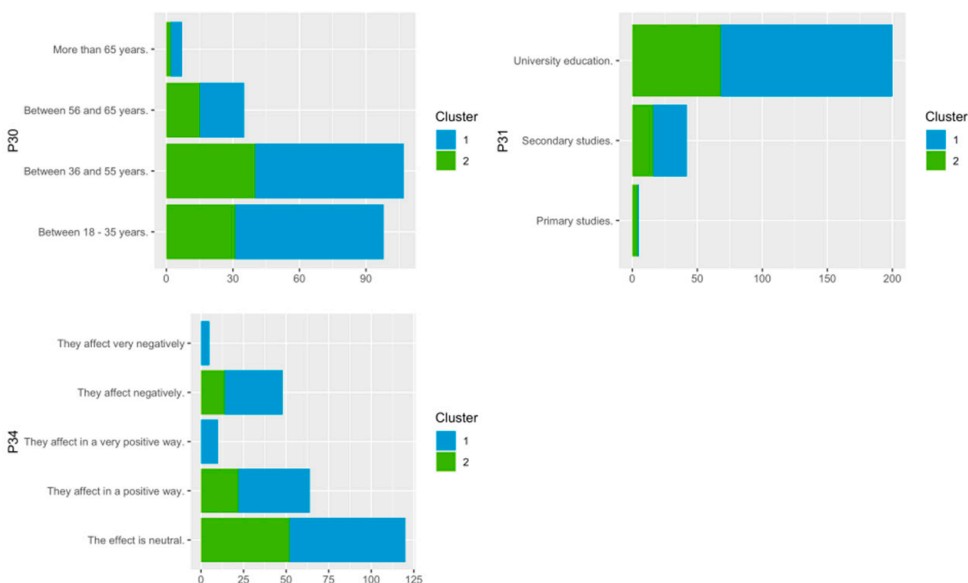

**Figure 6.** Summary of responses by cluster to questions with categorical responses from multiple categories. Source: compiled by the authors.

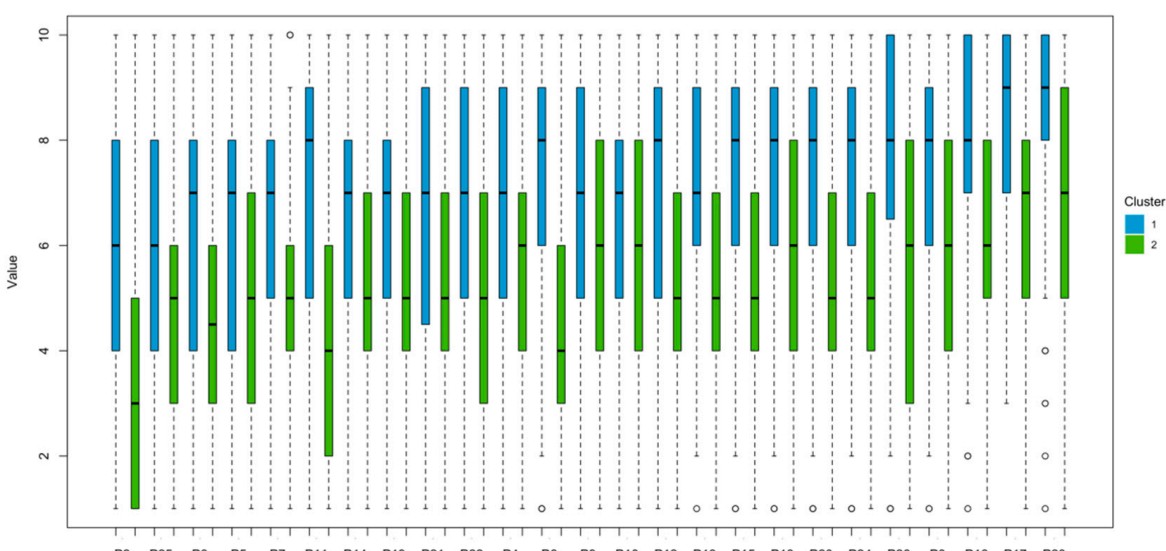

**Figure 7.** Summary of answers by cluster to the questions with quantitative answers (0–10). Source: compiled by the authors.

The vast majority of respondents from cluster 1 were 18 to 35 years old or 36 to 55 years old. Moreover, most of the respondents from Cluster 1 had a university education and were neutral or positive about the land use impact on their community.

The main finding of this research is presented in Figure 7, which shows how people who live in large cities value the different land use impact of tourism in their local communities more negatively than those that live in smaller towns. These results are more prominent for certain aspects, such as economic impact.

## 5. Conclusions

This study explored tourist perception of p2p platforms, and the land use impact associated with the consumption of these services in small and large cities. This work was based on two research questions. RQ1: Are there differences between the way tourists perceive the land use impact linked to these accommodations in small and large cities?

RQ2: Which are the effects perceived to a greater extent? Through these two questions, knowledge about the perception of consumers was broadened, which may have been conditioned by numerous factors.

In the existing literature, land use impacts on local communities associated with the activity of online platforms for intermediation of tourist accommodation have been described [9,11,25,27]. However, the consumption of hosting services mediated on p2p platforms, and the importance attributed by tourists to the land use impact in small and larger cities has not been analyzed. The findings indicated that the majority of respondents have a neutral and a positive opinion about the effect of tourism on their local environment. Only a few respondents suggested that tourism affected their local area negatively.

Of these, there were some factors highlighted as impacting negatively on their local community, in particular the increase in the price of housing and business establishments (rental and sale), the increase in prices in shops and bars, and high levels of congestion in public spaces with more traffic. This is in line with previous studies that suggest the negative land use impacts of tourism in local areas and contexts [5,13–15].

An interesting finding emerged, as most respondents pointed out that they were aware through the media of possible negative effects on the receiving communities associated with the expansion of tourist housing and only a few of the possible positive effects. Moreover, the majority of respondents perceived tourism as a great opportunity for local entrepreneurs. Evidence also revealed that there is a great difference between the perception of respondents on the land use impact of tourism in their local communities in large cities and small ones, with a worse perception of the use of p2p platforms by tourists that live in cities with a greater number of inhabitants. This is undoubtedly an added value of this research, as it allows us to discern the perception of this type of activity by the size of the city, which has not been explored before. Moreover, much broader research should be undertaken to explore in more depth the land use impact of p2p platforms in cities, for example exploring perceptions taking into account cultural differences within large cities in different countries. Further study could look at the perception of tourists exploring other variables, such as gender, age and education.

**Author Contributions:** Conceptualization, S.C. and A.M.; methodology, S.C. and A.M.; software, S.C., A.M., M.A.D.A. and G.M.M.d.C.; validation, S.C. and A.M.; formal analysis, S.C., A.M., M.A.D.A. and G.M.M.d.C.; investigation, S.C., A.M., M.A.D.A. and G.M.M.d.C.; resources, G.M.M.d.C.; data curation, M.A.D.A.; writing—original draft preparation, S.C. and A.M.; writing—review and editing, S.C., A.M., M.A.D.A. and G.M.M.d.C.; supervision, S.C., A.M., M.A.D.A. and G.M.M.d.C.; funding acquisition, G.M.M.d.C. All authors have read and agreed to the published version of the manuscript.

**Funding:** This paper is financed by Portuguese national funds through FCT—Fundação para a Ciência e a Tecnologia, I.P., project number UIDB/00685/2020.

**Data Availability Statement:** Data is unavailable due to ethical restrictions.

**Conflicts of Interest:** The authors declare no conflict of interest.

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
