# Peer review of "The Perception of the Impact of Land-Use on Small and Large Cities by Tourists Using p2p Platforms"

_land, doi:10.3390/land12040899_

Round 1

Reviewer 1 Report

Though the paper is technically sound, some detailed comments are listed as follows:

1.       In this paper, the research issue should be more clear.

2.       In the methodology, the question should be used as an attachment, not as the text of the paper.

3.       There are two same titles in the article: Cluster analysis

Author Response

Response to editors and reviewer comments

We would like to thank the editor and reviewers for their constructive comments on our paper and their very useful suggestions for revisions.

We have revised the paper to address fully these outstanding concerns. We have responded to the key issues raised concerning by the three reviewers. Full details of our response to each of the comments made are set out below and all changes made are highlighted in yellow and with track changes in the text.

If you require any further information, please do not hesitate to ask.

Yours sincerely,

Dr Sara Calvo

Reviewer 1 comments

Response

In this paper, the research issue should be more clear.

It has been made more clear. See highlighted in yellow.

 In the methodology, the question should be used as an attachment, not as the text of the paper.

Changed.

There are two same titles in the article: Cluster analysis

It has been modified. Highlighted in yellow Results.

Reviewer 2 Report

The paper is worth publishing.

I suggest excluding from the main text a detailed description (quoting) of the questions contained in the questionnaire (they take up a lot of space, .... regardless of this remark, question number 18 is even repeated) and posting the entire questionnaire as an attachment to the article.

I suggest that in the main text all questions be presented in full in some more aggregated form (e.g. when discussing research results).

The conclusions should no longer refer to the information given in the Introduction add in the methodological part (about the number of subjects and the circumstances of the study). There are still a few minor "repetitions" in the work, which may possibly be removed.

Author Response

We would like to thank the editor and reviewers for their constructive comments on our paper and their very useful suggestions for revisions.

We have revised the paper to address fully these outstanding concerns. We have responded to the key issues raised concerning by the three reviewers. Full details of our response to each of the comments made are set out below and all changes made are highlighted in yellow and with track changes in the text.

If you require any further information, please do not hesitate to ask.

Yours sincerely,

Dr Sara Calvo

Reviewer 2 comments

Response

The paper is worth publishing. I suggest excluding from the main text a detailed description (quoting) of the questions contained in the questionnaire (they take up a lot of space, .... regardless of this remark, question number 18 is even repeated) and posting the entire questionnaire as an attachment to the article.

Delete it and add it as an attachment to the article

I suggest that in the main text all questions be presented in full in some more aggregated form (e.g. when discussing research results).

Done.

The conclusions should no longer refer to the information given in the Introduction add in the methodological part (about the number of subjects and the circumstances of the study). There are still a few minor "repetitions" in the work, which may possibly be removed.

Done.

Reviewer 3 Report

The manuscript addresses a theme of interest to the tourists using p2p Platforms of cities. As tourists appear, the aspect of local community changes is an important issue in urban planning and land use. The manuscript however needs a very thorough revision on the following points:

1 - The abstract must be rewritten introducing the problem, the methodology, a brief discussion of the results achieved and the main findings.

2 - The introduction should be developed in such a way that the research question and the identified knowledge gap can be framed in a robust theoretical basis and in other papers already published on the subject, in particular the methodological approaches adopted.

3 - The material and methods section should be more understandable so that other researchers can replicate the methodology in other geographical contexts. 294 survey participants were surveyed over 2 years (2020~2022). Please clarify the reason for such a long time. Because of the lengthy investigation, it is necessary to explain that there is a possibility of different answers to the process of change.

4 – The reason for deriving issue questions should be announced.  

5 - The conclusions have to be focused on the findigns in an objective way. In addition, it is hoped that differentiation from existing studies and novelty will be presented in detail.

Author Response

We would like to thank the editor and reviewers for their constructive comments on our paper and their very useful suggestions for revisions.

We have revised the paper to address fully these outstanding concerns. We have responded to the key issues raised concerning by the three reviewers. Full details of our response to each of the comments made are set out below and all changes made are highlighted in yellow and with track changes in the text.

If you require any further information, please do not hesitate to ask.

Yours sincerely,

Dr Sara Calvo

Reviewer 3 Comments

Response

The abstract must be rewritten introducing the problem, the methodology, a brief discussion of the results achieved and the main findings.

Done.

The introduction should be developed in such a way that the research question and the identified knowledge gap can be framed in a robust theoretical basis and in other papers already published on the subject, in particular the methodological approaches adopted. 

Done.  

The material and methods section should be more understandable so that other researchers can replicate the methodology in other geographical contexts. 294 survey participants were surveyed over 2 years (2020~2022). Please clarify the reason for such a long time. Because of the lengthy investigation, it is necessary to explain that there is a possibility of different answers to the process of change.

Done. Apologies for this, data collection was taken during a year.

The reason for deriving issue questions should be announced.  

Done

The conclusions have to be focused on the findigns in an objective way. In addition, it is hoped that differentiation from existing studies and novelty will be presented in detail.

Done

Round 2

Reviewer 1 Report

I agree with the authors' revision.

Author Response

Response to editors and reviewer comments

We would like to thank the editor and reviewers for their constructive comments on our paper and their very final suggestions for revisions.

We have revised the paper to address fully these outstanding concerns. We have responded to the key issues raised concerning by the three reviewers. Full details of our response to each of the comments made are set out below and all changes made are highlighted in yellow and with track changes in the text. Proofreading has also been made for the article.

If you require any further information, please do not hesitate to ask.

Yours sincerely,

Dr Sara Calvo

Reviewer 2 comments

Response

I suggest that in the main text all questions be presented in full in some more aggregated form (e.g. when discussing research results).

It has been made more clear. See highlighted in yellow.

The conclusions should no longer refer to the information given in the Introduction add in the methodological part (about the number of subjects and the circumstances of the study). There are still a few minor "repetitions" in the work, which may possibly be removed.

Delete repetitions from conclusions and other sections in the paper.  

Reviewer 3 comments

Response

Firstly, please clearly define the small and larger cities as well as their scope in this paper.

Definition has been included in the paper.

Secondly, please add sufficient captions to the inserted images included in the paper.

Done.

Thirdly, what specific p2p platforms are being proposed in this paper? Please provide examples.

See highlighted in yellow.

Fourthly, the reference formatting should be modified to match the Land journal style.

See changes made highlighted in yellow.

Fifthly, as there are many papers that analyze these changes using social media data, it would be helpful to include recent research in the citation. For example, Kim, Seo, and Kwon (2021, Land Use Policy). This paper analyzed the perception of the Land Use impact on Small and Large Cities by tourists using p2p platforms. It is expected to have a significant impact on future follow-up studies.

Added paper. See in yellow.

Reviewer 3 Report

Your revised paper has faithfully reflected the comments from the previous review. Here are some additional comments. Firstly, please clearly define the small and larger cities as well as their scope in this paper. Secondly, please add sufficient captions to the inserted images included in the paper. Thirdly, what specific p2p platforms are being proposed in this paper? Please provide examples. Fourthly, the reference formatting should be modified to match the Land journal style. Fifthly, as there are many papers that analyze these changes using social media data, it would be helpful to include recent research in the citation. For example, Kim, Seo, and Kwon (2021, Land Use Policy). This paper analyzed the perception of the Land Use impact on Small and Large Cities by tourists using p2p platforms. It is expected to have a significant impact on future follow-up studies.

Author Response

(The authors gave the same response as above.)
